# Platelet-Based Liquid Biopsies through the Lens of Machine Learning

**DOI:** 10.3390/cancers15082336

**Published:** 2023-04-17

**Authors:** Sebastian Cygert, Krzysztof Pastuszak, Franciszek Górski, Michał Sieczczyński, Piotr Juszczyk, Antoni Rutkowski, Sebastian Lewalski, Robert Różański, Maksym Albin Jopek, Jacek Jassem, Andrzej Czyżewski, Thomas Wurdinger, Myron G. Best, Anna J. Żaczek, Anna Supernat

**Affiliations:** 1Department of Multimedia Systems, Faculty of Electronics, Telecommunication and Informatics, Gdansk University of Technology, 80-233 Gdańsk, Poland; cygerts@gmail.com (S.C.);; 2Ideas NCBR, 00-801 Warsaw, Poland; 3Department of Algorithms and System Modeling, Faculty of Electronics, Telecommunication and Informatics, Gdansk University of Technology, 80-233 Gdańsk, Poland; 4Laboratory of Translational Oncology, Intercollegiate Faculty of Biotechnology, Medical University of Gdańsk, 80-210 Gdańsk, Poland; 5Center of Biostatistics and Bioinformatics, Medical University of Gdańsk, 80-210 Gdańsk, Poland; 6Independent Researcher, 80-211 Gdańsk, Poland; 7Department of Oncology and Radiotherapy, Medical University of Gdańsk, 80-210 Gdańsk, Poland; 8Department of Neurosurgery, Amsterdam University Medical Center, 1081 Amsterdam, The Netherlands

**Keywords:** liquid biopsy, machine learning, interpretability, robustness, RNA sequencing

## Abstract

**Simple Summary:**

Liquid biopsies are a non-invasive way to diagnose and monitor cancer using blood tests. Machine learning can help understand the genetic data from these tests, but it is challenging to validate clinical applications. In our study, we first compiled a large-scale dataset for cancer classification. Then, we extracted relevant features from the data and performed a binary classification, with the prediction outcome of either a sample collected from a cancer patient or a sample collected from an asymptomatic control. We used different convolutional neural networks (CNNs) and boosting methods to evaluate the classification performance. We have obtained an impressive result of 0.96 area under the curve. Finally, we tested the robustness of the models using test data from novel hospitals and performed data inspection to find the most relevant features for the prediction. Our work proves the great potential of using liquid biopsies for cancer patient classification.

**Abstract:**

Liquid biopsies offer minimally invasive diagnosis and monitoring of cancer disease. This biosource is often analyzed using sequencing, which generates highly complex data that can be used using machine learning tools. Nevertheless, validating the clinical applications of such methods is challenging. It requires: (a) using data from many patients; (b) verifying potential bias concerning sample collection; and (c) adding interpretability to the model. In this work, we have used RNA sequencing data of tumor-educated platelets (TEPs) and performed a binary classification (cancer vs. no-cancer). First, we compiled a large-scale dataset with more than a thousand donors. Further, we used different convolutional neural networks (CNNs) and boosting methods to evaluate the classifier performance. We have obtained an impressive result of 0.96 area under the curve. We then identified different clusters of splice variants using expert knowledge from the Kyoto Encyclopedia of Genes and Genomes (KEGG). Employing boosting algorithms, we identified the features with the highest predictive power. Finally, we tested the robustness of the models using test data from novel hospitals. Notably, we did not observe any decrease in model performance. Our work proves the great potential of using TEP data for cancer patient classification and opens the avenue for profound cancer diagnostics.

## 1. Introduction

In recent years, medicine has made numerous attempts to involve machine learning (ML) algorithms to improve patient outcomes and facilitate the work of clinicians. In addition to improving patients’ quality of life, machine learning can be used to reduce doctors’ professional burnout [1]. Primary uses of machine learning are related to image processing [2], but artificial intelligence (AI) in general also allows scientists to keep up with the multitude of data generated for patients: MRI scans, CT scans, radiotherapy results, histology slides, sequencing results, and others. Such formats of data can be classified using versatile methods, including image recognition [3] or, more recently, graph recognition [4].

Precision medicine has long sought to leverage molecular information about the disease to improve patient outcomes [5]. Widely used tissue biopsy samples and imaging methods are limited by constraints on geographical availability, sampling frequency, and incomplete disease representation. Hence, the medical world’s attention has turned to minimally invasive liquid biopsies, which enable the interrogation of bodily fluid components: DNA, RNA, proteins, or even whole cells [6]. The potential of liquid biopsies is highlighted by studies that show blood-based assays can track even the evolutionary dynamics and heterogeneity of the disease, detecting the very early emergence of therapy resistance, residual disease, and recurrence [5]. Offered by liquid biopsies, minimally invasive sample collection enables the detection and characterization of multiple diseases in a convenient, repeatable, and real-time manner [3,7,8]. The developed modern tests rely on: (a) tremendous progress in our ability to purify and analyze body fluid components (such as cells, platelets, DNA, RNA, protein, or metabolite biomarkers) and (b) the introduction of high throughput techniques that provide unprecedented resolution of the analysis. As liquid biopsies are bound to transform patient care in the coming years, it is essential to emphasize that the current challenge is to make them a standard clinical tool [7]. The generated data complexity enforces the need for more advanced models than assuming a simple cut-off for the final result interpretation [9].

As liquid biopsies introduce a high level of data complexity, applying machine learning to data processing becomes a natural direction. The number of studies combining liquid biopsy analysis and machine learning is continuously expanding. The most recent applications of machine learning are related to circulating tumor cell (CTC) enumeration and CTC imaging [10,11] or small RNA profiling with the use of principal component analysis [12]. Machine learning has also been used for classification based on DNA mutational profiles, combined with data augmentation [12] and protein analysis [13]. Regarding platelet RNA sequencing data in cancer, so far, this type of data has been analyzed with the use of either particle swarm optimization-enhanced support vector machines [14,15] or deep neural networks [3].

Applying machine learning methods in clinical applications is still very challenging [16]. For example, during deployment, the model performance may drop drastically due to the effect of the model learning to explore spurious features from the training set, which may not generalize to novel data (e.g., from external hospitals) [17]. Other challenges include model stability, performance in the underrepresented classes [18], and robustness, which constitute essential research topics for machine learning in general [19]. Therefore, the performance obtained on internal datasets can only be treated as an upper boundary of the system’s performance in the real world, and using external data is crucial for validation.

Previously, we established that the platelet transcriptome retains remarkable stability in healthy individuals but changes dramatically under the influence of disease [20]. Since then, we have created a deep-neural-network-based tool that allows for ovarian cancer detection. This unique tool uses the biological knowledge deposited in the Kyoto Encyclopedia of Genes and Genomes (KEGG) and converts RNA sequencing data collected from patients’ blood to images [3]. We also studied the utility of tumor-educated platelets and circulating tumor DNA for preoperative endometrial cancer diagnosis, including histology determination. We provided evidence that liquid biopsy can complement, if not replace, standard microscopic traditional biopsy evaluation [21]. The aforementioned studies lay the groundwork for a novel approach to liquid biopsies.

However, before the full potential of utilizing data from the RNA sequencing of platelets is explored, we need to thoroughly test the stability and performance of the proposed methods. In this work, we capitalize on the ability of machine learning to unearth unapparent signals hidden in liquid biopsy data. Moreover, we take further advantage of the introduction of biological structures: the representation of various types of biological data as networks rather than as single features [22]. An overview of our approach is summarized in Figure 1. In brief, we collected liquid biopsy data from multiple medical units and investigated the binary, simple classification of cancer versus non-cancer samples consisting of 720 cancer cases and 422 non-cancer cases. Boosting algorithms were then used to find the most important features, which allowed us to gain insight into the model decisions. Finally, held-out data from novel locations (unexposed during training) were used to test the models’ robustness. In summary, the contributions of this paper are as follows:We present a comparative study of various machine learning algorithms (convolutional neural networks (CNNs), boosting) on the task of liquid biopsy cancer classification using a recently introduced novel feature vector extraction.We show that using knowledge from the KEGG database works as efficient feature preselection.We study the robustness of the presented algorithms when presented with the samples collected at hospital locations that were not used in the training process.We identify the most important features for classification.

## 2. Data

### 2.1. Datasets

We gathered data from several datasets. However, it is essential to emphasize that all platelet samples were processed strictly according to the guidelines published by Best et al. [23]. The dataset GSE156902 [8,24] contained cohorts of asymptomatic controls, multiple sclerosis patients, selected as a group of patients with a non-malignant disease with a significant inflammatory component, and patients with different forms of cancer. GSE158508 [3] contained cases of ovarian cancer patients, GSE184904 included endometrial cancer patients [21], and GSE89843 included non-small cell lung cancer (NSCLC) patients [15]. The dataset of sarcoma was previously published [25]. The authors of the original study supplied the raw counts. Several samples were excluded in the process of quality control based on poor data quality or clinical status with additional confounding factors.

Healthy controls, NSCLC, and sarcoma samples were collected at VU University Medical Center (Amsterdam, The Netherlands), Netherlands Cancer Institute (Amsterdam, The Netherlands), and Massachusetts General Hospital (Boston, MA, USA).

Samples of endometrial and ovarian cancer patients were collected at the Department of Gynecology, Gynecological Oncology, and Gynecological Endocrinology at the Medical University of Gdansk (MUG). The study was approved by the Independent Ethics Committee of the Medical University of Gdansk (NKBBN/434/2017). All patients from all included hospitals signed informed consent forms. Procedures involving human subjects were in accordance with the Helsinki Declaration, as revised in 1983.

The final dataset contained the data of patients with 6 different cancer types, patients with brain metastasis from different primary sites, and 422 non-cancer patients (asymptomatic controls and multiple sclerosis patients). An overview of the data is presented in Table 1. The data were aggregated and normalized together. Each of the constituent datasets underwent the same preprocessing.

### 2.2. Data Preparation

The DESeq2 package [26] with a variance stabilizing transform [27] was used for the normalization of the data. The data was then annotated using the Gencode v19 GRCh37 annotation [28]. The samples were then subjected to quality control, namely, samples with less than 100,000 total reads were excluded from further analysis. Only splice variants that could be mapped to a transcript with Gencode status “known” were included. If two IDs were mapped to the same gene name, expression data for the ID marked in the Gencode general transfer format (GTF) as Level 1 was used. Normalized and filtered expression profiles were then the basis for constructing images, where the color of each pixel corresponded to the expression level of a certain gene [3].

As each RNA-sequenced platelet sample consisted of reads belonging to 39,843 different splice variant types, we decided to experiment with CNNs and convert these transcripts to a two-dimensional array. The value in each position represents the number of transcript counts detected; hence, the higher the gene expression of each splice variant, the higher its value. Values in the array were arranged according to their biological significance. As our primary classifier was developed to recognize the tumor-educated platelets (TEPs) of cancer patients, we decided to focus especially on pathways that might be deregulated due to tumor development. Hence, we searched the KEGG database [29] and selected signaling pathways corresponding to four crucial aspects: cancer, metabolism, signaling processes, and the immune system. Combining these four groups of pathways resulted in a higher accuracy than using just pathways marked as directly related to cancer [3]. R package GAGE was used to gather the KEGG pathway data [30]. In each pathway, KEGG IDs that were not linked to the expression level of any particular gene were removed from the corresponding rows.

The full vector obtained from each platelet sample contained 267 rows corresponding to the 267 signaling pathways. In general, the rows can be divided into the following groups: metabolism, genetic information processing, environmental information processing, cellular processes, immune systems, other organismal systems, cancer, and other human diseases.

### 2.3. Dataset Split by Location

Machine learning models are known to be vulnerable to various confounding factors. They exhibit reduced classification performance when presented with data from novel hospitals [17]. Thus, we repeated the experiments using a new data split to simulate the truly independent test set with samples from different hospitals. The samples used to train the model were collected at VU University Medical Center, Amsterdam, The Netherlands (707), and the Medical University of Gdansk (67). Samples for the new independent test set were collected at other hospitals: Massachusetts General Hospital, USA (87), Netherlands Cancer Institute, Amsterdam, The Netherlands (192), Radboud University Medical Center Nijmegen (4), and Medical University of Vienna (5). Patients with sarcoma were not included in this particular analysis, as any split based on their location of origin would result in an even more imbalanced split of the other samples. Furthermore, unlike ovarian or endometrial cancer cases present only in the training set, sarcomas are of embryological origin, different from other types of tumors. Details of the data split are presented in Table 2.

## 3. Models

In this section, three models used for the classification of cancer versus not-cancer are briefly presented. The code and data will be made available to the public upon paper acceptance.

### 3.1. imPlatelet

The first model used for the experiments was the classifier that was previously shown to obtain very high accuracy in the discrimination between healthy donors and ovarian cancer [3]. A deep neural network model was built using the Keras R package with TensorFlow backend [31]. It consists of 10 layers, including 8 hidden layers: 2 two-dimensional convolutional layers, each with four filters and a kernel size of 3 × 3, 4 densely connected layers with a gradually reduced number of units, and 2 dropout layers. Binary cross-entropy was used as a loss function, and gradient optimization was performed using the adadelta algorithm. Since the dataset was imbalanced, the classes received weights proportionate to their frequencies, resulting in a measure of “balanced accuracy” used in the experiments for all the presented models.

### 3.2. Standard CNN

We also experimented with standard CNN models. We choose ResNet architecture [32] with 18- and 34-layer variants. As we found no significant difference between the two models, a smaller variant was used (because of the small dataset size). CNN allowed us to obtain a much smaller model than imPlatelet (11.2 M of parameters instead of 145.3 M). Standard ResNet implementation from the PyTorch library was used [33]. Further, the dropout layer [34] was added before the last layer, with a dropout probability of 0.2. The Stochastic Gradient Descent optimizer was used with a learning rate of 0.1, decreasing every seven epochs by one order of magnitude and with a weight decay of 0.001. Binary cross-entropy was used as a loss function, with weight-balanced loss (similar to the previous model).

### 3.3. Gradient Boosting

Another algorithm tested on our dataset was the eXtreme (XGBoost) classifier [35]. We conducted a random hyperparameter search. The optimized hyperparameters included learning rate, maximal tree depth, number of estimators, and dropout rate (of the input features). Throughout a few trials, ranges of more promising values were extracted, and during the final experiments, 150 chosen settings were tested three times. Each training round that tested 150 settings took approximately 11 h on i7 7700HQ. Additionally, the early stopping round hyperparameter was set to 15 to reduce overfitting on the training set.

## 4. Experiments

### 4.1. Model Comparison

The test set included 30% of the stratified random samples. The remaining 70% of the samples were used for stratified 5-fold cross-validation. Class balance was preserved in each subset, including the split of controls into healthy donors used in ovarian cancer classification. Each model was then tested using the test set. Unless stated otherwise, each experiment was repeated three times. For data augmentation, we also experimented with MixUp [36], but no significant improvements over the baseline were found.

Table 3 presents the obtained results. Somewhat surprisingly, all the methods yielded very similar results. Notably, ResNet-18 obtained the same accuracy, as large as the imPlatelet classifier, using a significantly smaller number of parameters. An important finding was that the boosting algorithm achieved competitive accuracy, but its use is all the more justified as it allows for the interpretability of the model decisions. All the models achieved impressive, very high results (approximately 0.96 AUC in the test set).

#### 4.1.1. Use of KEGG Expert Knowledge

Based on our previous experiments, we decided it would be interesting to compare the accuracy of DNNs and CNNs as CNNs utilize local information. To verify whether the local information is useful in this task (that is, grouping the related rows and having each signaling pathway in a different row in the array), we conducted experiments on random permutations of rows and columns (Table 4). We demonstrated that the local information had a negligible impact on the performance of CNNs as the balanced accuracy scores on validation and test sets were only slightly degraded. Consequently, using the information from the KEGG pathways (and grouping related pathways together) did not enhance classifier performance by providing a biological background.

Hence comes the question of whether using the KEGG database is useful at all for this task. Using the KEGG pathway database allowed us to remove a number of features, from over 39,000 variables to roughly 23,000, eliminating noise and focusing on domain knowledge; however, it was not known what the impact of such filtering on classifier performance would be. As such, we additionally trained a boosting algorithm on all of the features (39,000 variables) and observed that the balanced accuracy and AUC dropped to 0.819 and 0.922, respectively (compared to 0.889 and 0.96 when using KEGG feature preselection).

#### 4.1.2. Robustness Test

Finally, we performed a robustness test by testing data from samples collected from a different location than the data used for the training process (as presented in Section 2.3), without any adaptation of finetuning, to imitate real-world application. The obtained results are presented in Table 5. The ROC curves are presented in Figure 2. Impressively, no performance drop is observed when transferring the data obtained at the new hospital. Although the bias associated with different collection points cannot be entirely excluded, the possibly introduced confounding factor did not significantly affect the classification.

### 4.2. Feature Importance

In this section, we explore the groups of KEGG-arranged features used by the models. In the first experiment, we intended to understand the impact of different groups on the final model accuracy. Hence, we used only one of the groups from the KEGG database and verified the final classifier accuracy (Figure 3). No striking differences were found between the analyzed groups. The lowest accuracy was obtained when using the KEGG information concerning the genetic information processing group, but the test balanced accuracy remained relatively high (0.818). Using the information from only some of the groups performed almost as well as when all the data were used; for example, using the cancer information group allowed us to achieve 0.87 of balanced accuracy, whereas the model relying on all the data achieved only slightly higher accuracy of 0.89. Therefore, we concluded that the information needed by the classifier to make a decision was already available within each of the groups used in the KEGG database.

Next, we utilized the explainability of the boosting approach. An XGBoost model was trained on the entire dataset, and the feature importance was extracted from it. Then, a series of experiments was conducted to determine the performance of the test set when using only a limited number of features for the training (from 10 to 500). For each experimental setting, the models were trained for 100 training rounds, each with a different hyperparameter set. Figure 3 shows the obtained results for a limited number of features. Apparently, when using only 10 features, the model already reached 90% in the area under the curve (AUC) metric. The major increase happened up to 100 features, when approximately 0.955 of the AUC was obtained. The ROC curve is depicted in Figure 4.

The top 10 splice variants implicated in class prediction included genes associated with cell signaling and gene expression regulation (NCOA4, PTPN6), ribosome formation (RPL7A, RPS25, RPS18, RPL10), immune system (HLADRA, HLA-F, CD27), and cell energetics (NDUFB11). Platelets are known to play a significant role in inflammation and immune responses. Thrombocytosis, an increased platelet count, is very common among cancer patients. Platelet-cancer crosstalk generates a vicious feedback loop: tumor cells secrete molecules that activate platelets, promoting, in turn, cancer-associated inflammation, cell proliferation, dissemination, and immune system evasion. Hence, the decrease in RNA expression in platelets collected from cancer patients stems from the intense translation of the mentioned genes to proteins in response to different cues associated with disease progression [37].

According to a gene expression profiling study published by Gnatenko et al., NCOA4 encodes cDNA for RFG (RETproto-oncogene RET/PTC3) and belongs to the top 50 human platelet-expressed genes [38]. Its role in platelet functioning has not been studied in detail. By contrast, PTPN6 encodes a protein belonging to the tyrosine phosphatase (PTP) family. PTPs are known signaling molecules. They regulate multiple cellular processes, such as cell growth, differentiation, the mitotic cycle, or oncogenic transformation. PTP-1B, a member of this family, has been reported to be an essential positive regulator of thrombus formation [39]. Furthermore, the highly expressed PTPN6 gene in immune cells has been correlated with a favorable prognosis in gastric cancer. Both PTPN6 and CD27 are considered subjects for immunotherapy [40,41]. The vast repertoire of RNAs is carried over into mature platelets, along with the functional spliceosome system and ribosomes, which explains the transcripts related to ribosome formation [42]. As cancer cells rewire cell functioning, increasing their biosynthetic and metabolic activities, a complex and highly energy-consuming process occurs [38]. This explains the necessity for the ribosome biogenesis reflected in platelets.

### 4.3. Discussion

Contrary to what we have published previously [3] and contrary to the related works involving machine learning [43], the addition of KEGG pathways [44] did not improve the accuracy of classification in this study, as shown in the experiment when using random permutations of rows and columns for a CNN-based classifier. However, the KEGG-based preselection of features proved to improve boosting classifier accuracy. Moreover, the experimental data showed that the classifiers were already relatively accurate when using data from the most important features. The obtained accuracy of the order of approximately 90% could allow for the performance of much cheaper screening tests based on liquid biopsies [45]. This is imperative in the context of potential future clinical applications of the classifier. Determining a small subset of features that provide sufficient accuracy would allow the use of reverse transcriptase quantitative polymerase chain reaction (RT-qPCR) instead of RNA sequencing in the laboratory. The RT-qPCR technique is faster, cost-effective, and readily available compared to the latter. We also showed that our model is highly accurate and robust. Hence, the location of the material collection did not significantly affect the prediction effectiveness.

Platelets demonstrate certain benefits over other liquid biopsy sources: convenient isolation, abundance, high quality of extracted nucleic acids, and the ability to process RNA information in response to external cellular cues [46]. Hence, combining RNA-sequenced TEPs with multiple machine learning approaches warrants opportunities for future biomarker trove discovery, paving the way toward optimal, personalized diagnostic strategies.

## 5. Conclusions

In this work, we have presented a meaningful, large-scale comparison of different methods for liquid biopsy classification in the form of RNA-sequenced platelets collected from a cohort of cancer patients and non-cancer donors.

We have shown that using the information from the KEGG database as a feature preselection allows us to improve classifier performance. The features belonging to various KEGG pathways have high predictive power, and even separate groups of pathways could provide high accuracy for classification on their own.

Further, we have validated that the models work well when using data from novel hospitals, which is of great importance for clinical trials. We conclude that the boosting method seems to be the optimal selection; it achieved the same accuracy as the more computationally expensive CNN-based methods. Furthermore, it allowed us to add a level of interpretability to the model, extracting the most important features that affected the final prediction results. Using only the 20 top features was sufficient to obtain 0.924 in the AUC metric. Further development and implementation of machine learning applied to platelets in clinical settings will mandate solid interdisciplinary measures. We have established features (splice variants) that are crucial for robust classification, enabling the development of a highly sensitive assay that is time- and cost-effective.

## Figures and Tables

**Figure 1 cancers-15-02336-f001:**
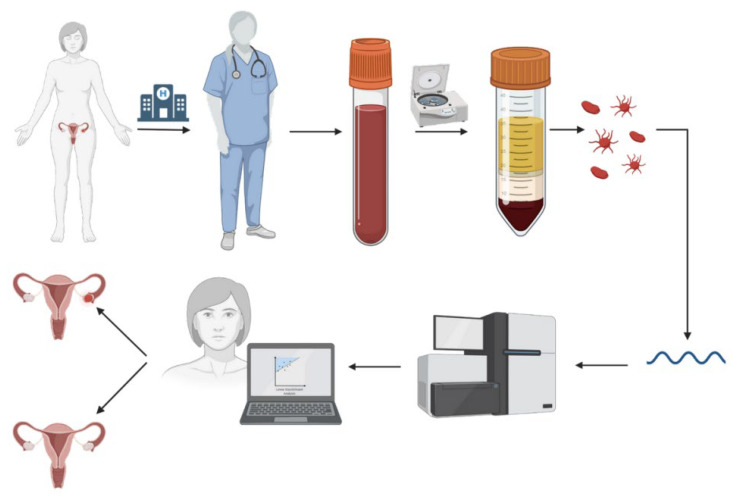
General workflow of sample collection and further classification. First, a patient treated at the hospital is enrolled in the study based on the inclusion criteria specified by clinicians; next, a blood sample is collected, and platelets are processed, RNA is extracted from platelets and sequenced. Then, bioinformatics processing follows, and a machine learning model is used for sample classification (healthy donor versus patient with diagnosed cancer).

**Figure 2 cancers-15-02336-f002:**
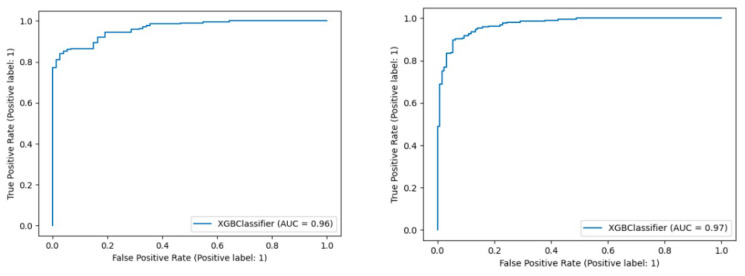
Area under the curve for the first experiment (left image) and on the transfer to the new hospital (right image).

**Figure 3 cancers-15-02336-f003:**
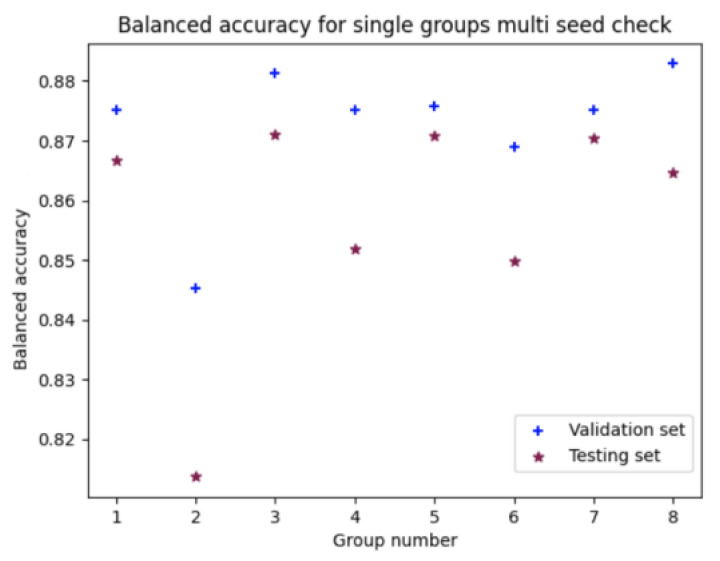
Accuracy of the imPlatelet model when trained only on one of the groups from the KEGG database. The consecutive groups correspond to: (1) metabolism, (2) genetic information processing, (3) environmental information processing, (4) cellular processes, (5) immune system, (6) other organismal systems, (7) cancer, and (8) other human diseases.

**Figure 4 cancers-15-02336-f004:**
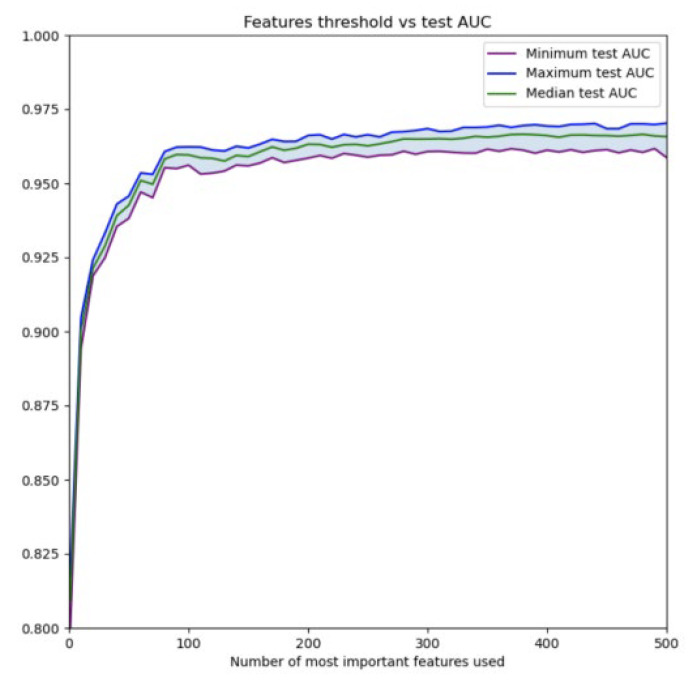
AUC of the Boosting method using the most important features.

**Table 1 cancers-15-02336-t001:** Collected data statistics.

-	EC	OC	NSCLC	GBM	Brain Metastasis	Sarcoma	Asymptomatic Controls	Multiple Sclerosis
Num patients	39	28	329	234	51	39	338	84

**Table 2 cancers-15-02336-t002:** Dataset split for the transfer to a new hospital.

-	EC	OC	NSCLC	GBM	Brain Metastasis	Sarcoma	Asymptomatic Controls	Multiple Sclerosis
Training	39	28	142	215	25	Not included	260	65
Test	0	0	185	4	26	Not included	54	19
Total	39	28	327	219	51	Not included	314	84

**Table 3 cancers-15-02336-t003:** Results of evaluated models.

Model	Val Bal. Acc.	Test Bal. Acc.	Val AUC	Test AUC
imPlatelet	0.902	0.891	0.970	0.966
ResNet-18	0.898	0.883	0.957	0.950
Boosting	0.907	0.889	0.962	0.960

**Table 4 cancers-15-02336-t004:** Effect of permuting rows and columns of the images on the model performance.

Model	Val AUC	Test AUC
ResNet-18	0.957	0.950
Permuted rows	0.959	0.951
Permuted columns	0.955	0.945

**Table 5 cancers-15-02336-t005:** Results of the experiment transfer to a new hospital.

Model	Val Bal. Acc	Test Bal. Acc	Val AUC	Test AUC
imPlatelet	0.898	0.854	0.970	0.966
ResNet-18	0.913	0.857	0.965	0.958
Boosting	0.909	0.878	0.967	0.953

## Data Availability

The data presented in this study are openly available in NCBI GEO, reference number GSE156902 (asymptomatic controls, multiple sclerosis patients, cancer patients), GSE158508 (ovarian cancer patients), GSE184904 (endometrial cancer patients), and GSE89843 (non-small cell lung cancer patients).

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
