# Peer review of "Platelet-Based Liquid Biopsies through the Lens of Machine Learning"

_cancers, 2023, doi:10.3390/cancers15082336_

Round 1

Reviewer 1 Report

I can hardly find the flaw in this manuscript, it is superbly written, with sound design mimicking the real world settings, attaining to stable and well described methodologies and SOPs, appropriate control cohorts, smart combination/comparison of three independent classifiers and relevant conclusion on possible extraction of several transcripts constituting plausible RT-PCR panel for validation in similar cohorts. This is basically only thing I would wish to see besides what is described in the paper. 

The only missing information is the description of cancer patients whose samples are included in the study. Are they collected/enrolled in a screening scenario? at the first diagnosis? what is the staging? are they homogeneous or not? Is there any relation between accuracy of the algorithm (already high) and tumor type/stage'?  This would be the information and comment I would like to see addressed

Otherwise ready to publish

Author Response

Comment #1: I can hardly find the flaw in this manuscript, it is superbly written, with sound design mimicking the real world settings, attaining to stable and well described methodologies and SOPs, appropriate control cohorts, smart combination/comparison of three independent classifiers and relevant conclusion on possible extraction of several transcripts constituting plausible RT-PCR panel for validation in similar cohorts. This is basically only thing I would wish to see besides what is described in the paper.

The only missing information is the description of cancer patients whose samples are included in the study. Are they collected/enrolled in a screening scenario? at the first diagnosis? what is the staging? are they homogeneous or not? Is there any relation between accuracy of the algorithm (already high) and tumor type/stage'? This would be the information and comment I would like to see addressed.

Otherwise ready to publish

Response #1: The blood samples were collected from patients at the time of the diagnosis at collaborating hospitals. Importantly, the samples were taken before the treatment started, hence eliminating one of the potential confounding factors. The staging was performed by clinicians, according to the official staging guidelines. For gynecologic malignancies, FIGO staging was used. The patient group was not homogeneous, but the distribution of age and sex was consistent with age and sex distribution at the time of diagnosis.

The accuracy of detection of non-small cell lung cancer was the highest, but the differences were relatively small, e.g. for the hospital transfer split, the overall AUC on the test set was above 93%. When restricted to non-malignant donors and either NSCLC or GBM patients, the AUC was close to 95% and 92%, respectively. The only group that performed a bit worse was the brain metastasis group, which was also noticeably smaller and more heterogeneous (patients had primary tumors of different types and locations). The study cohort consisted of patients at all stages. While the metastatic disease was overrepresented in the NSCLC cohort, accuracy remained high also for the less advanced stages.

Reviewer 2 Report

Cygert et al. used RNA-sequencing and machine learning to differentiate cancer cell from tumor cells. The work is very interesting since blood platelets, a neglected source of tumor cell information, is not considered as a key cell for liquid biopsy. However, I have few questions:

1.       For me was not clear if your model could be used for detection of cancers in different stages;

2.       Usually, model generated from one cancer cohort may not be applicable to another one, did you have variation between the tumor types analysed?

3.       Did you try to combine your model with other clinical/molecular characteristics to discriminate cancer patients from healthy controls?

4.       Was patient and control blood samples collected and processed under the same conditions? If not did you found variations?

5.       Did you run any validation assay for the genes you found (q-PCR),?

Author Response

Comment #1: For me was not clear if your model could be used for detection of cancers in different stages.

Response #1: The study cohort consisted of patients at all stages. While the metastatic disease was overrepresented in the NSCLC cohort, accuracy remained high also for the less advanced stages.

Comment #2: Usually, model generated from one cancer cohort may not be applicable to another one, did you have variation between the tumor types analysed?

Response #2: The accuracy of detection of non-small cell lung cancer was the highest, but the differences were relatively small, e.g. for the hospital transfer split, the overall AUC on the test set was above 93%. When restricted to non-malignant donors and either NSCLC or GBM patients, the AUC was close to 95% and 92%, respectively. The only group that performed a bit worse was the brain metastasis group, which was also noticeably smaller and more heterogeneous (patients had primary tumors of different types and locations).

Comment #3: Did you try to combine your model with other clinical/molecular characteristics to discriminate cancer patients from healthy controls?

Response #3: Unfortunately, for some of the patients' additional data on the potentially relevant biomarkers was missing, hence we decided to focus strictly on the platelet expression profiles.

Comment #4: Was patient and control blood samples collected and processed under the same conditions? If not did you found variations?

Response #4: The samples were collected following the same protocol, as specified by the authors of [https://www.nature.com/articles/s41596-019-0139-5 ]. The samples were collected, preprocessed and frozen under the same conditions, at hospitals where they were gathered. Frozen samples were then sent to VUMC and sequenced in the same laboratory, following the same protocol.

Comment #5: Did you run any validation assay for the genes you found (q-PCR)?

Response #5: Since the study was retrospective, a validation on the same cohort was not feasible. However, we did publish the results of RT-qPCR validation experiments in the hereby paper: https://pubmed.ncbi.nlm.nih.gov/36905391/ RT-qPCR validation of the newly collected Gdańsk cohort is also underway.